# Non-spherical particles in optical tweezers: A numerical solution

**Joonas Herranen** [1]*, **Johannes Markkanen**[2], **Gorden Videen**[3,4,5], **Karri Muinonen**[1,6]

**1** Department of Physics, University of Helsinki, Finland, **2** Max Planck Institute for Solar System Research, Göttingen, Germany, **3** Army Research Laboratory, Adelphi, Maryland, United States of America, **4** Space Science Institute, Boulder, Colorado, United States of America, **5** Kyung Hee University, Gyeonggi-do, South Korea, **6** Finnish Geospatial Research Institute FGI, National Land Survey, Finland

* joonas.herranen@iki.fi

## Abstract

We present numerical methods for modeling the dynamics of arbitrarily shaped particles trapped within optical tweezers, which improve the predictive power of numerical simulations for practical use. We study the dependence of trapping on the shape and size of particles in a single continuous wave beam setup. We also consider the implications of different particle compositions, beam types and media. The major result of the study is that for different irregular particle shapes, a range of beam powers generally leads to trapping. The trapping power range depends on whether the particle can be characterized as elongated or flattened, and the range is also limited by Brownian forces.

## 1 Introduction

Optical tweezers have been under close study in the last decades, after the pioneering results of the recent Nobel laureate A. Ashkin [1]. Since then, optical tweezers have been subject to a multitude of experimental studies [2]. Modeling optical tweezers is, needless to say, an important part of fundamental understanding of the device and its possibilities. In a vast majority of studies, only the forces and torques applied by the tweezers are considered [3]. This is due to the fact that the dynamical modeling of the tweezers considers constantly changing particle position and orientation. This in turn requires the solution of a new scattering problem. This is a daunting task for traditional methods such as geometric optics or any scattering solution in which incident field properties are not separable from the scattering problem.

Currently, toolboxes to solve for more limited cases of optical tweezers do exist. Existing frameworks deal with both geometric optics [4], and particles sized near or at the Mie scattering regime making use of particle symmetry properties in the $T$-matrix method [5], which is the semianalytical generalization of Mie scattering. In this work, we present an application of the $T$-matrix method for the modeling of dynamics of arbitrarily shaped particles, which is a recent breakthrough of the $T$-matrix method in general. The question at hand has been identified as a primary open question related to the simulation of optical tweezers [6].

In previous studies [7, 8] we have combined into a single software framework a $T$-matrix [9] scattering solution for particles of arbitrary shape and composition with an integrator of

**Data Availability Statement:** All relevant data are within the paper and its Supporting Information files.

**Funding:** KM was supported by Finnish Academy of Sciences grant SA 1298137 (https://www.aka.fi/

en/). The funders had no role in study design, data collection and analysis, decision to publish, or preparation of the manuscript.

the equations of motion of the particle. This dynamical response by electromagnetic interactions is hereafter called scattering dynamics. The methodology presented is based on an efficient and numerically exact solution of the volume integral equation for inhomogeneous particles [10], which imposes no constraints on the scatterer shape.

The current study extends the previously developed dynamic light-scattering framework for modeling optical tweezers, `scadyn` (S1 Software). In order to achieve this, we first need to consider the addition of other beam shapes than the plane wave as the source of illumination. Second, the physical effects in an optical tweezers environment, such as drag, need to be considered. In particular, we focus on the category of beam modes known as Laguerre-Gaussian (LG) modes in three different media: vacuum, air, and water.

In the following, we introduce a model for numerical studies of optical tweezers model for arbitrarily shaped rigid particles. Such setups have been shown experimentally to have unexpected, possibly shape- and composition-dependent behavior under illumination [11]. We demonstrate the numerical method by studying the dynamical response of cubic, ellipsoidal, and non-spherical, so-called Gaussian random ellipsoidal [12], particles under different radiation environments. We confirm that the trap efficiencies are a relatively effortless method of predicting the shape-dependent behavior of particles in different optical tweezers, especially when the drag effects are weak. We also simulate possible measurements of both particle position and angular frequency utilizing only the tweezers themselves.

## 2 Theory

In this work, we use the full scattering solution of non-spherical particles of arbitrary composition to perform optical-tweezers modeling. In this section, we briefly review the foundational light-scattering framework, fully described by Herranen et al. [7], and introduce the physical model of the optical tweezers system used in the calculations in the following.

### 2.1 Particle modeling

The particles considered in this work are composed as rigid tetrahedral meshes, with each tetrahedron having a constant index of refraction and density. Larger particles can be modeled more efficiently as aggregates [10] or using geometric optics, although these approaches are not explored in this work.

In this work, we consider non-spherical particles in the regime of Mie scattering, where neither the Rayleigh approximation nor the assumptions of geometric optics are valid. As the set of non-spherical particles, cubic and different deformed ellipsoidal (as Gaussian random ellipsoids [12]) particle shapes are considered, exemplified in Fig 1.

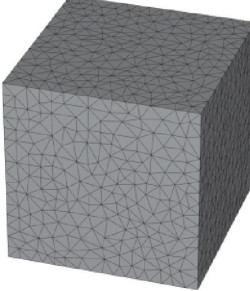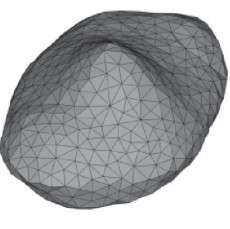

**Fig 1. Example particle shapes considered in this work.** The Gaussian random ellipsoid (right), which is based on a normal ellipsoid with axis ratios 1:0.8:0.4, has a standard deviation of 0.125 and a correlation length 0.35.

In optical tweezers, where the beam power is focused to an area of a few square microns, the local intensity, or radiant flux, will easily reach values as high as $10^{10}$ W/m². Thus, to avoid over-heating, it is often preferable for trapped materials to have low absorptivity, which is characterized through the imaginary part of the optical constant. The particle material for the majority of this work is assumed to be quartz-like, with an optical constant $1.4588 + i9.783 \times 10^{-5}$ at wavelength $\lambda = 1064$ nm, and density $\rho = 2300$ kg/m³. Some other materials, a completely non-absorbing (diamond-like, $2.4 + i0$), an absorbing opaque ($1.75 + i0.3$), and a metallic-like absorbing ($0.25 + i7.5$), will be compared in a separate subsection.

## 2.2 Incident electromagnetic field modeling

In optical tweezers, the beam shape is an important variable when considering the dynamics due to scattering, or scattering dynamics. In the vector spherical wave function (VSWF) expansion, the beam shape coefficients of the chosen shape must be found first. While straight-forward in principle, proper care must be taken when comparing numerics and real experiments. Particularly, the localized approximations, that are often used for modeling LG beams, need to be adapted carefully [13]. In this work, we do not address this issue further and consider azimuthal LG beam modes, constructed as in the Optical Tweezers Toolbox [14], near the beam focus in the paraxial approximation [15].

The $LG_{pl}$-beam, with its modes labeled by the radial index $p$ and the azimuthal index $l$, is of particular interest in optical trapping. The $LG_{00}$-mode has a simple Gaussian intensity profile, illustrated in the top half of Fig 2 for linearly and circularly polarized cases. The $LG_{0l}$-modes, exemplified in the bottom half of Fig 2, consist of torus-like intensity profiles along which the trapped particles can move. In general, the LG-beams are described in [16]. The amplitude profile of a cylindrically symmetric $LG_{0l}$-mode is given by

$$E_{0l}(r, z) = w(z)^{-1} \exp\left(-\frac{r^2}{w(z)^2}\right)\left(\frac{\sqrt{2}r}{w(z)}\right)^{|l|}, \qquad (1)$$

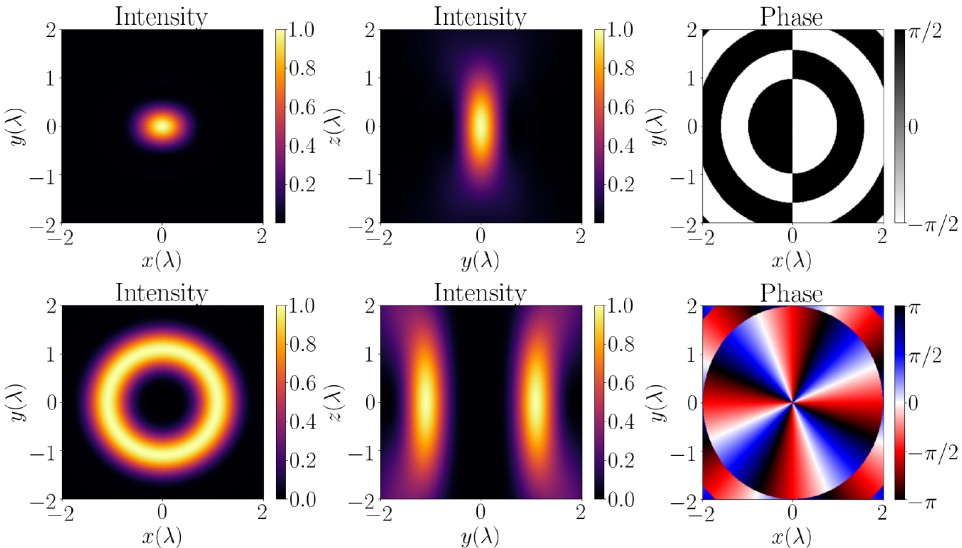

**Fig 2.** Top: The normalized intensity profile in the (x,y) and (x,z) planes, and the (x,y) phase profile of a linearly polarized $LG_{00}$, or Gaussian mode, reconstructed from the vector spherical wave function expansion. Bottom: The same for a circularly polarized $LG_{03}$ mode. In both cases, all planes pass through the origin, numerical aperture NA = 0.8, and the medium is air.

where phases and normalization factors have been omitted, $r$ is the distance from origin, and $w(z)$ is the beam width at height $z$.

The LG-modes are highly localized, thus they are usable in optical trapping applications. In the middle column of Fig 2, the cross sections near the focal plane are shown. In canonical optical tweezers studies, two types of forces, scattering and gradient force, are usually considered separately both for more intuitive picture of the tweezers and to compare directly the non-trapping and trapping effects. The full scattering solution used in this work to analyse dynamical responses cannot differentiate between these two forces. This fact stands for any consideration of non-spherical particles, as currently analytic decompositions of forces exist only for spherical particles in the Mie size regime [17]. Thus, only the total force due to interaction with light can be considered.

## 2.3 Tweezers based on scattering dynamics

We construct a model of optical tweezers with a perfectly rigid particle suspended in vacuum by the tweezers themselves. In more realistic applications, multiple effects, such as flow dynamics [18] and non-rigid dynamics [19, 20], can have major roles. The model presented in this work accounts for one common realization of optical tweezers: a rigid particle in a medium such as air or water, suspended by the tweezers against gravity.

The scattering dynamics are solved by integrating the particles' equations of motion with the symplectic Lie-Verlet method [21], which has proven to be an efficient and easily implementable method of integrating Euler's equations of rotation. Translational equations in this work are updated using Euler's method.

When considering classical optical tweezers theory, interactions with light are expected to dictate the movement of small particles. However, as the particles move in optical traps or tweezers, they are subjected to other forces as well. The forces are gravity, flow-dynamical forces such as drag, and the Brownian force. Collisions with other particles of the same size are not considered. This constraint is fundamentally due to the inability to handle interacting scatterers i.e. multiple particles in the scattering dynamics solution. Basic flow dynamical results for spherical particles are well-known in the form of Newton and Stokes' drags. For non-spherical particles, many of these effects are either difficult or impossible to solve without numerical methods. It is still crucial to account for these forces, as the inertia of the medium is the only non-electrodynamic means of countering the scattering force of the tweezers. Hence, the flow dynamical effects in this work are accounted as order-of-magnitude estimates by considering analytic spherical particle results of either Newton drag or Stokes' drag depending on the Reynolds number of the relative flow.

The optical forces and torques are obtained from the total electric and magnetic fields, as the momentum-transferring interactions between light and the particle, described originally by Maxwell [22] and Poynting [23]. The momentum transfer can be solved analytically in the VSWF expansion, thus the forces and torques are essentially functions of the total fields [7, 24, 25]. Formally, this solution can be represented via force and torque efficiencies $\mathbf{Q}_F$ and $\mathbf{Q}_N$, which are defined as

$$
\begin{cases}
\mathbf{F}_{\text{scat}} = \dfrac{\pi a^2}{2} \epsilon E^2 \mathbf{Q}_F, \\[2mm]
\mathbf{N}_{\text{scat}} = \dfrac{\pi a^2}{2k} \epsilon E^2 \mathbf{Q}_N,
\end{cases}
\tag{2}
$$

where $a$, $k$, $\epsilon$, and $E$ are the equivalent sphere radius of the particle, wave number, permittivity of the medium, and the incident field amplitude, respectively.

As stated above, the medium causes both rotational and translational drag on the particle. In optical tweezers, small particles rarely obtain velocities that result in large Reynolds numbers Re = $\rho_{\text{med}} va/\mu$, where $\rho_{\text{med}}$ is the density of the surrounding medium, $v$ the relative speed of the liquid with respect to the particle and $\mu$ the dynamic viscosity of the medium. The low Reynolds number results from typical maximum velocities of order mm/s from to optical torques. In this work, we approximate $\mu \approx 10^{-3}$ Pa · s for water and $10^{-5}$ Pa · s for air. Using these values, velocities of the order 10 m/s are needed for large values of Re. However, when Re is large, in the case of untypical conditions and media, the translational drag is the Newton drag,

$$\mathbf{F}_D = -\frac{C_D}{2}\pi a^2 \rho_{\text{med}} v^2 \hat{\mathbf{v}},$$ (3)

where $C_D$ is the dimensionless shape-dependent drag coefficient. In practical calculations, we assume $C_D = 1$ for simplicity. For low values of Re, which applies for most of the situations in this work, the usual Stokes' drag of the form

$$\mathbf{F}_D = -6\pi\mu a\mathbf{v},$$ (4)

is used.

For non-spherical particles, the rotational mode dictates the rotational drag, as can be phenomenologically understood by considering different possible rotational states of e.g. a cylinder. This shape-dependent drag is disregarded for practical reasons in this work. Rotational drag is also dependent on the rotational Reynolds number $\text{Re}_\omega = \rho_{\text{med}}\omega a^2/\mu$, and for a sphere in the high $\text{Re}_\omega$ case it can be written as [26]:

$$\mathbf{N}_D = -\frac{C_\omega}{2}\rho_{\text{med}}a^5\omega^2\hat{\boldsymbol{\omega}},$$ (5)

where the rotational drag coefficient $C_\omega$ is set to unity.

Compared to translational speeds, angular speeds are high and can vary very much, as is demonstrated in later sections. However, $\text{Re}_\omega$ is likely to have very low values as well, when considering small particles. This is the situation especially in water. Again, when $\text{Re}_{\omega\rightarrow}$ 0, rotational drag is in Stokes' regime, given by Faxén's second law [27], for a sphere,

$$\mathbf{N}_D = -8\pi\mu a^3\boldsymbol{\omega}.$$ (6)

In total, the forces and torques acting on a particle with volume $V$ and density $\rho$ in optical tweezers are given by

$$\begin{cases} \mathbf{F} = \mathbf{F}_{\text{scat}} + \mathbf{F}_D + \mathbf{F}_G + \mathbf{F}_m + \mathbf{F}_M, \\ \mathbf{N} = \mathbf{N}_{\text{scat}} + \mathbf{N}_D. \end{cases}$$ (7)

Above, $\mathbf{F}_G = (\rho - \rho_{\text{med}})V\,\mathbf{g}$ is the gravitational force, $\mathbf{F}_m = 1/2\rho_{\text{med}}\,V\mathrm{d}\mathbf{v}/\mathrm{d}t$ is the added mass force and $\mathbf{F}_M = C_M\rho_{\text{med}}V(\boldsymbol{\omega}\times\mathbf{v})$ the lateral Magnus force [28], with $C_M = 1$.

**2.3.1 Brownian forces.** The random motion due to collisions with the molecules in the surrounding medium have been disregarded in the above discussion. The application of the Brownian forces is simple in the context of this work, as this exact problem has been thoroughly discussed [29]. The Brownian force is accounted for by the Langevinian addition in temperature $T$ to the total force, resulting in ballistic inertial motion, given by

$$\mathbf{F}_B = \sqrt{2k_B T\gamma}\mathbf{W},$$ (8)

where $\gamma = 6\pi\mu R$ is the friction coefficient of a sphere with a radius $R$, and $\mathbf{W}$ is a random force

vector with normally distributed components, zero mean, and variance $1/\Delta t$. In practice, the integration time step $\Delta t$ defines the Brownian variance.

Brownian torques, which are here assumed to have a negligible effect on the particle drift, as the beams themselves are able to produce angular speeds up to thousands revolutions per second even with drags slowing the particles. Nevertheless, the torques would be constructed heuristically as

$$\mathbf{N}_B = \mathbf{r}_B \times \mathbf{F}_B, \qquad\qquad (9)$$

where $\mathbf{r}_B$ is a randomly distributed impact distance from the origin between zero and the maximum radius of the particle in the plane perpendicular to the force spanning the cross section of the particle.

In this work, we mostly analyse the stability of the modeled tweezers, and as such do not add any randomizing forces by default. However, as in real systems that are not in-vacuo, Brownian effects have to be taken into account when performing any comparison between experiments and models. Thus, in section 3.3, we will also demonstrate and discuss the implications of Brownian forces.

## 3 Simulation of optical tweezers

The dynamical response of an arbitrary particle is difficult both to foresee and illustrate. As such, some basic measures of the dynamical response are illuminating. We will show that the case of optical traps, trap efficiency, defined as the force efficiency as a function of position, is one particularly useful predictor for non-spherical particles. Furthermore, we will show that the scattered intensity from a focused beam provides means of measuring the position and angular frequency of the particle.

Trap efficiency is a core quantity in optical tweezers literature. An optical trap is often modeled as a set of spring constants, as the force efficiencies as a function of position tend to form a downward slope near the zeros of the force efficiency components.

Non-spherical particles are often helical, meaning that when they scatter light, right- and left-handed circular polarizations of incident light behave differently. Thus a net torque would appear, even when circularly polarized beams are not considered. In the absence of significant rotational drag, non-spherical particles can reach large angular velocities.

For this reason, in the following analysis, rotationally averaged trap efficiencies are used. From the infinite amount of possible rotational axes, a natural choice would be one of the principal axes, set parallel to the beam propagation direction. In the left side of Fig 3, the trap efficiency of a cube rotating about its only unique principal axis in a vacuum is shown. In the example, a quartz-like cube has a radius $a = 0.2$ $\mu$m of an equivalent volume sphere, which corresponds to an edge length 0.231 $\mu$m and a size parameter $ka = 2\pi a/\lambda \approx 1.2$. The incident beam is a linearly $+x$-polarized $LG_{00}$ beam with numerical aperture NA = 0.8 and amplitude 1 MV/m at the $z = 0$ intensity maximum, corresponding to a total power of 0.4 mW. It should be noted, that if the incident beam is circularly polarized, trap efficiencies in the transverse $x$- and $y$-directions will average to be equal.

The zeros of the trap efficiency indicate that the cube has an equilibrium point around $(x, y, z) \approx (0, 0, 0.8)$ $\mu$m. Solving the dynamical problem explicitly for the cube initially at origin, suspended in air, and affected by gravity reveals a similar response to one predictable from the trap efficiency. The addition of gravity drags the equilibrium point down, as is illustrated in the right side of Fig 3. The cube will almost immediately set around $(x, y, z) \approx (0, 0, 0.75)$ $\mu$m. Generally, the final position of the particle will also depend on the specific shape and focusing of the beam.

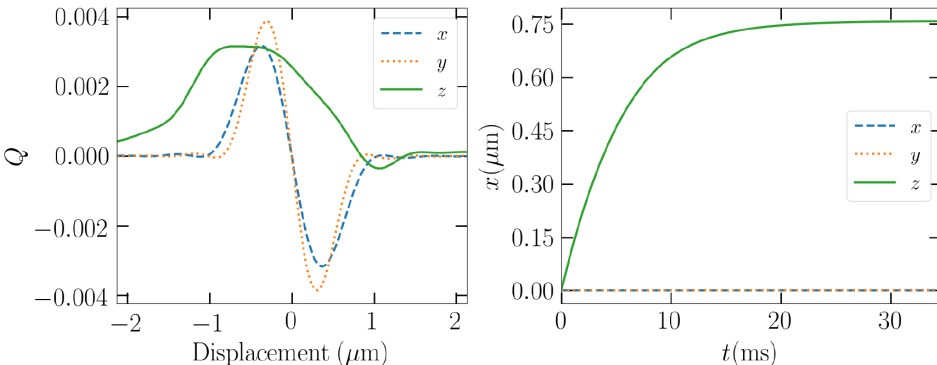

**Fig 3.** Left: Rotationally averaged linearly polarized $LG_{00}$ trap efficiency components for a 0.2 $\mu$m cube spinning about its principal axis, which is parallel to the beam propagation direction $+z$. Right: Position as a function of time when suspended in air by the optical trap.

The ability of this kind of a rudimentary beam setup to trap particles will thus depend on whether or not there exists a negative slope across $Q = 0$ in the $z$-direction. Otherwise, the particle will bounce on the trap due to the beam and gravity, the stability depends largely on the true spinning state. This means, that an erratically spinning particle will most likely be blown too high and with so much $x$- or $y$-directed velocity that it will be out of the reach of the trap. To have the maximum potential of the particle to be able to reach a stable mode of rotation before they are slung out of the trap, we hereafter consider circularly polarized beams.

### 3.1 Different media

The medium in which the particle is submerged will induce drags on the particle and have an effect on the trap efficiency due to changing relative refractive index. The integration scheme is based on solving the fundamental law of dynamics, and thus always includes the inertial forces on the particle. However, as the drags constrain the particle to move at the terminal velocity, inertial forces are often disregarded. Thus, in media with different viscosities, where drags dominate, only time scale differences are expected.

For the integration to be numerically stable, adaptive time stepping is used. The time steps are given by a maximum allowed revolution, displacement, and angular velocity change in a single update. In this case, when the time step is dependent on both the optical forces and torques and current velocities, the beam power used becomes an important choice. This is due to the fact that in a more viscous medium with too small beam power the particle reaches its terminal velocity very quickly, yet near the beam focus the time steps are still forced to be very small. If the time steps were not very small, the drag forces in a simple integration scheme would almost certainly numerically overshoot and velocity would not converge to the terminal velocity. The problem vanishes when terminal velocity is assumed to be reached instantly, and the particle position and orientation are updated directly. However, later we note that especially the evolution of angular velocity is of such importance that it is still worth the effort to update the full equations of motion.

Next, in Fig 4, we compare the behaviour of the cubic shape in vacuum, air, and water in a circularly polarized trap. In these numerical tests, the beam power of 230 mW is usually taken. However, in some cases, such as now in vacuum, where drag forces and torques are completely disregarded, manual fine tuning is necessary to be able to trap the particle even adequately. In practice, the optimal beam power is such that the upward optical force is just larger than

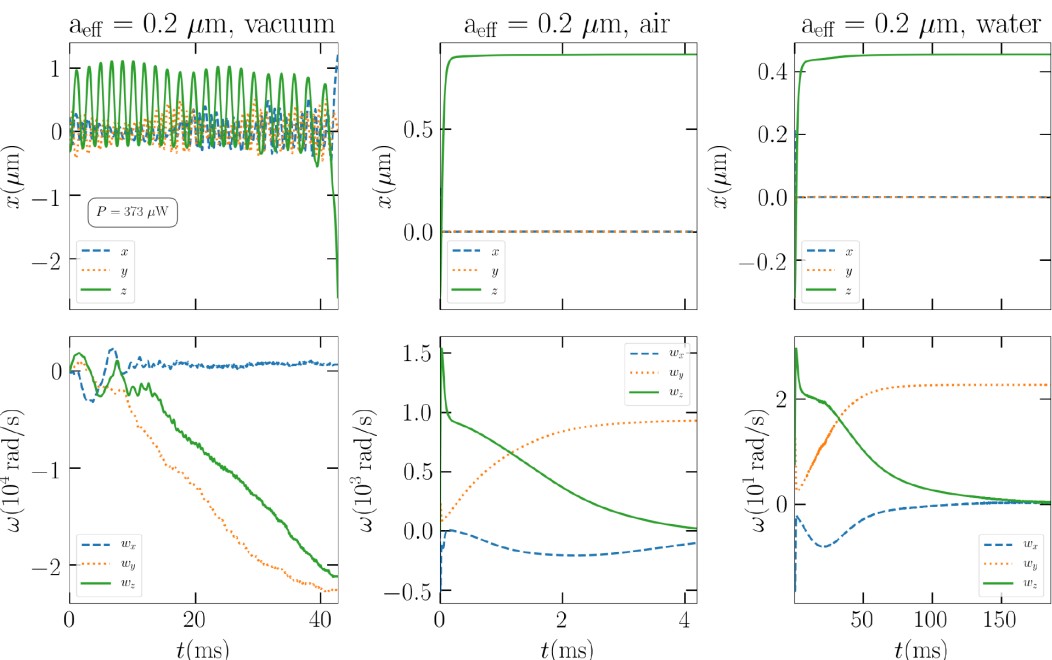

**Fig 4. The $a_{eff} = 0.2 \, \mu$m cube in the LG$_{00}$ trap immersed in vacuum (left), air (middle), and water (right).** The beam powers are 373 $\mu$W, 230 mW, and 230 mW, respectively.

gravity. We also see that the resulting particle motion is highly different from the cases of air and water, which mostly differ between their time scales. This is again due to the absence of drags.

The actual dynamical response of the cube is seen to be similar with only time scale differences depending on the medium. However, the trap efficiency is affected by the medium, as we see for the cases of differently sized cubes and Gaussian ellipsoids in Fig 5. The shape of the trap efficiency is nearly unaffected in other cases than of the 1-$\mu$m Gaussian particle, for which the transverse trap efficiency slightly changes. The change in medium results in different magnitude and scaling of the trap efficiency. For example, the change in the equilibrium $z$-position of the 0.2-$\mu$m cube correlates to the new location of the zero of the $z$-component of the trap efficiency. The change in the trap efficiency magnitude has a size-dependent response when immersed in water instead of air. For smaller particles, trapping efficiency increases when the relative refractive index increases, while for the large 1 micron particles the magnitude decreases.

Then, we focus on the dynamics of a Gaussian ellipsoid in vacuum in three different cases, where $a$ = 0.1, 0.2, and 1 $\mu$m. Now, in Fig 6, we see that the trap efficiencies can be used to predict the instability of the trap for this particular shape. However, rotations that differ from those used in the calculation of averaged trap efficiencies bring additional complications. In the leftmost column of Fig 6, the 0.1 $\mu$m particle is expected to be trapped according to the trap efficiencies. Indeed, the particle stays in the trap for some while, but we see that eventually the rotational motion changes (see smaller, zoomed out images in Fig 6) so that the particle is ejected. In the other cases, where the trap efficiencies do not indicate successful trapping, the particles are promptly ejected from the trap.

When suspended in air, however, the different sized Gaussian ellipsoids of all sizes stay in the trap. Drag forces can be attributed in keeping the particle near the beam focus and in the

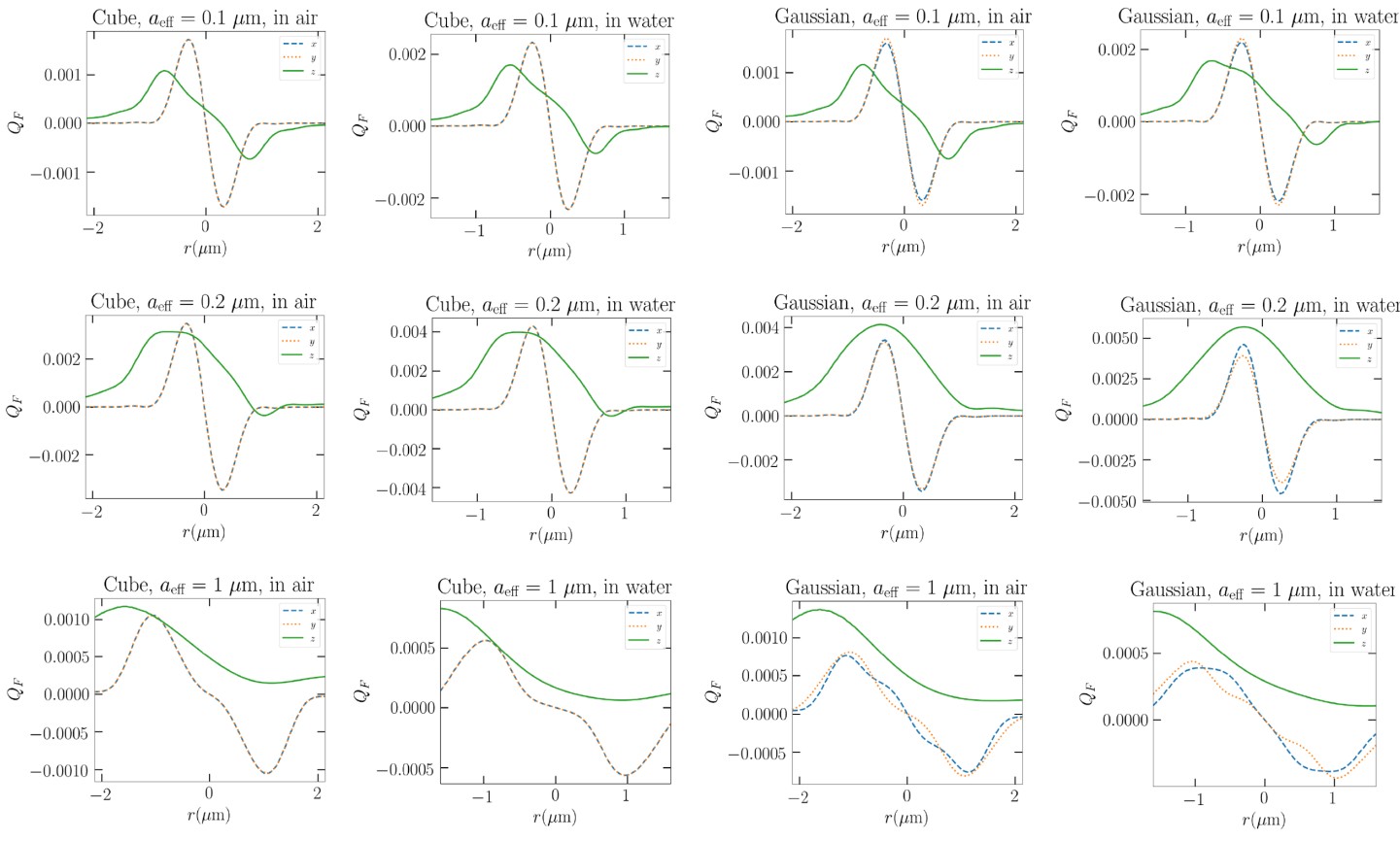

**Fig 5. Trap efficiencies of different sized and Gaussian and cube particles in both water and air.**

dissipation of excess velocity, should the particle be ejected. In Fig 7, the trapped positions for the three different particle sizes are presented. In the case of the largest particle, the 230 mW beam proves to be enough to result in large oscillations, where the trap can just pull the particle back to the focus. As the beam power is lowered to 4.57 mW, the particle will eventually rotate around the trap at a stable height, as seen in the fourth panel of Fig 7.

To illustrate the effect of beam power and irregularity to the stability of the particle $z$-position, we consider the $a_{\text{eff}} = 0.2\ \mu$m cube and Gaussian ellipsoid with a variable power of an LG$_{00}$, again circularly polarized. The mean $z$-positions of 250 separate runs are collected in Fig 8. The means are calculated disregarding the initial settling phase by automatic cut-off. For a significant power range, the cube behaves like the cube in linearly polarized beam in Fig 4, and it settles to about $z = 0.85\ \mu$m. Similarly, the Gaussian ellipsoid settles most of the time like the specific example in the second panel of Fig 6. However, after the beam power reaches 10 W, the particle is quickly ejected from the beam. Comparison with the symmetric case of the cube, which is trapped by up to an order of magnitude more powerful beam, this either means that the deformed particle does not have enough time to settle in the trap, or it cannot do so at all due to its shape, before it is out of the reach of the trap.

## 3.2 Different shapes

In the previous section, we saw that the irregular shape of the particle induces irregular rotations, which make the particle response to optical tweezers much more difficult to predict by

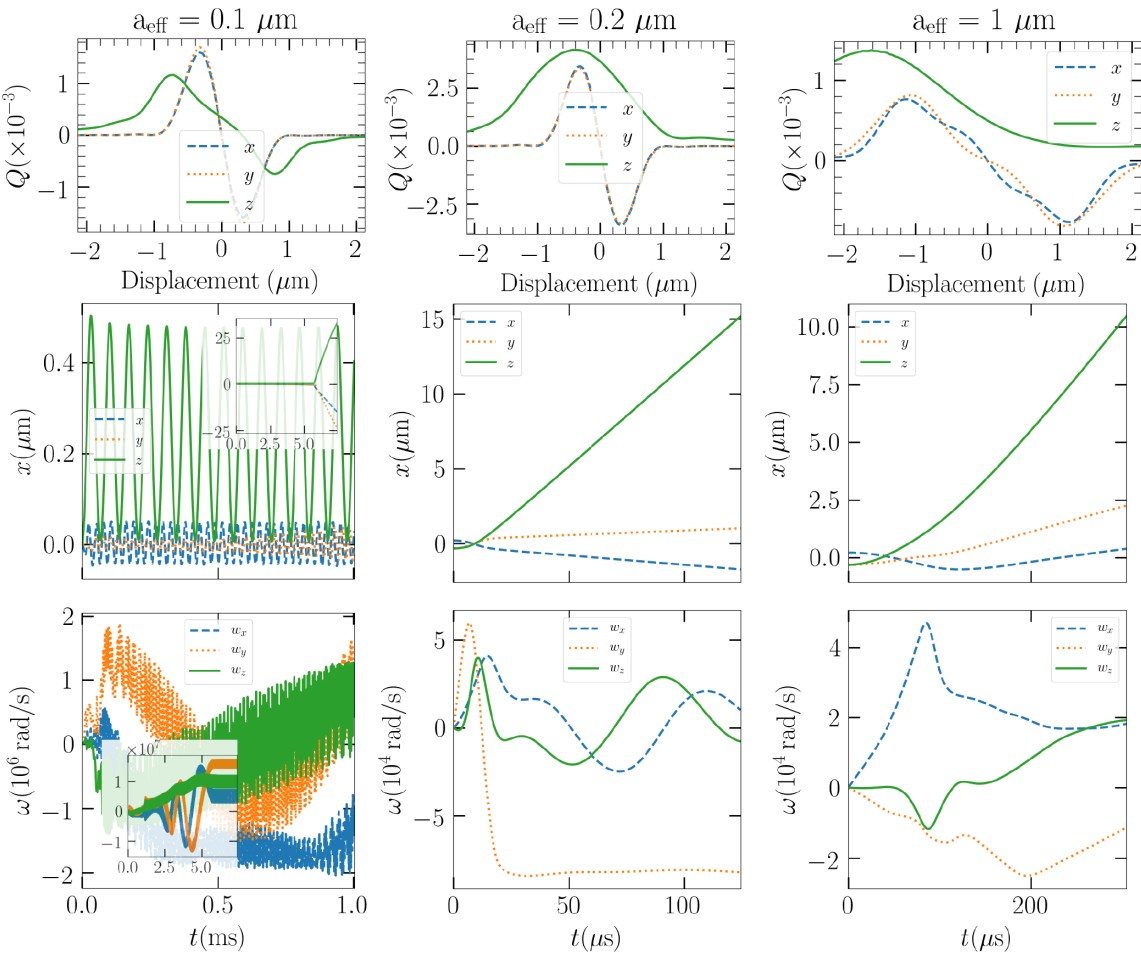

**Fig 6.** Top row, the trap efficiencies averaged over a revolution about the major principal axis as a function of displacement along the laboratory axes for the Gaussian ellipsoid of sizes $a_{\text{eff}}$ = 0.1, 0.2, and 1 $\mu$m, respectively. Middle, the position of the particles in vacuum as a function of time in the LG$_{00}$ optical trap. Bottom, the angular velocities.

simple methods. Next, we systematically grow the amount of available shapes and study their response in the tweezers.

We start by considering spheroidal base shapes, with axis ratios $a$:$b$:$c$ of 2:2:5, 3:3:5, 4:4:5, 5:5:5, 5:5:4, 5:5:3, and 5:5:2. Then, we perform the random Gaussian deformations with the

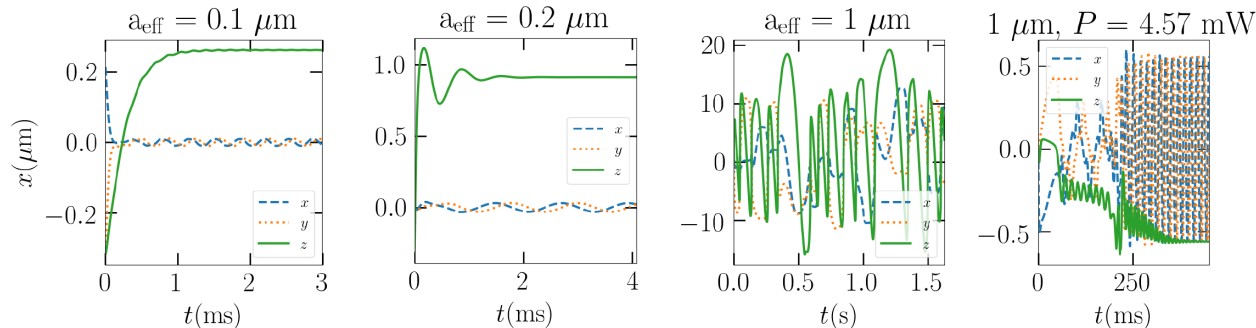

**Fig 7. From left to right: The positions of the $a$ = 0.1, 0.2, and 1 $\mu$m Gaussian ellipsoids in air as a function of time in the LG$_{00}$ optical trap.** The first 3 panels correspond to beam power 230 mW, and the last panel to 4.57 mW.

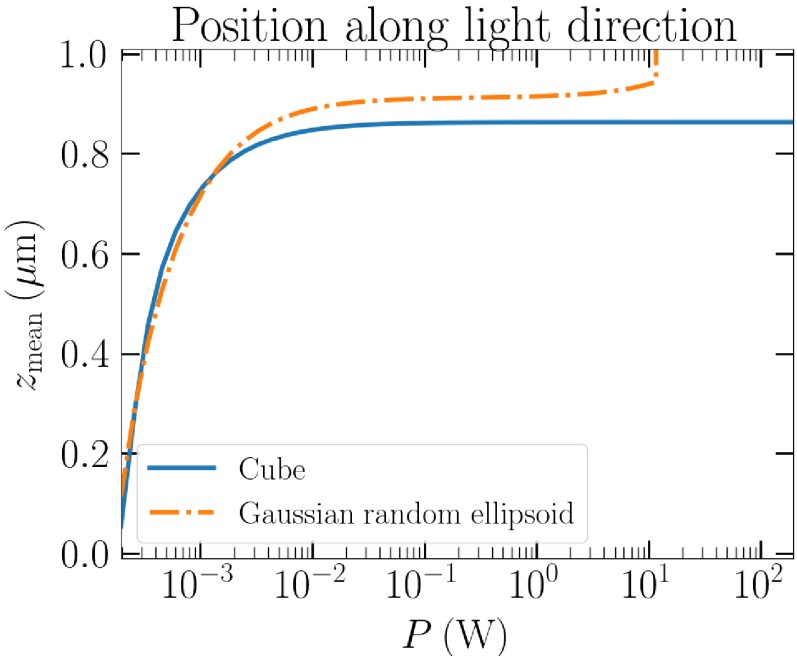

**Fig 8. The mean $z$-position of the cube and the Gaussian ellipsoid as a function of beam power.**

same standard deviation and correlation length as for the example ellipsoid, namely $\sigma = 0.125$ and $l = 0.35$. The resulting shapes are presented in Fig 9. As it has already become quite clear that the deformed shape has significantly different scattering, and thus dynamical response, it should also be highlighted that the deformation has an effect on the particle inertial properties, too. The right panel of Fig 9 presents what kind of smooth ellipsoid axis ratios would be needed to produce an undeformed set that has the same principal moments of inertia as the Gaussian spheroid sample.

Proceeding similarly as at the end of the last section, we calculate the mean $z$-position of this Gaussian spheroidal sample as a function of beam power, for two sizes, $a_{\text{eff}} = 0.2$ and 1

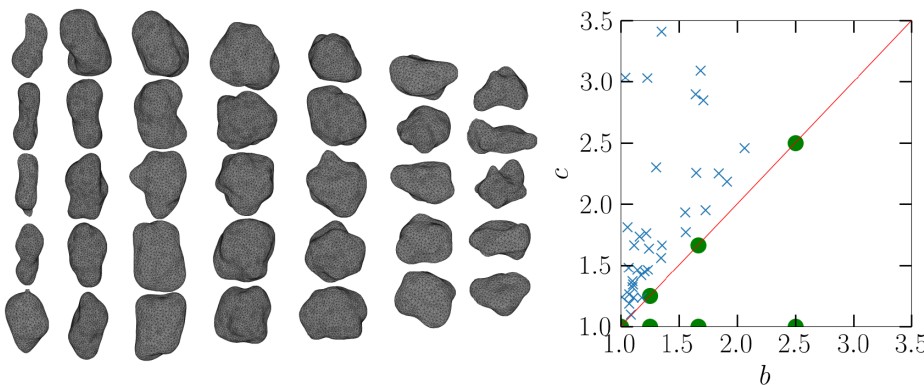

**Fig 9. The Gaussian spheroid sample (left) and (right) axis ratios of ellipsoids with the same principal inertial properties as of the Gaussian spheroid sample.** The dots represent the undeformed base shapes, which are only 7, and crosses the axis ratios needed to sample. The axis ratios are normalized so that $a = 1$. All values lie in reality in the upper left triangle, and the dots are mirrored to the other side for visual effect.

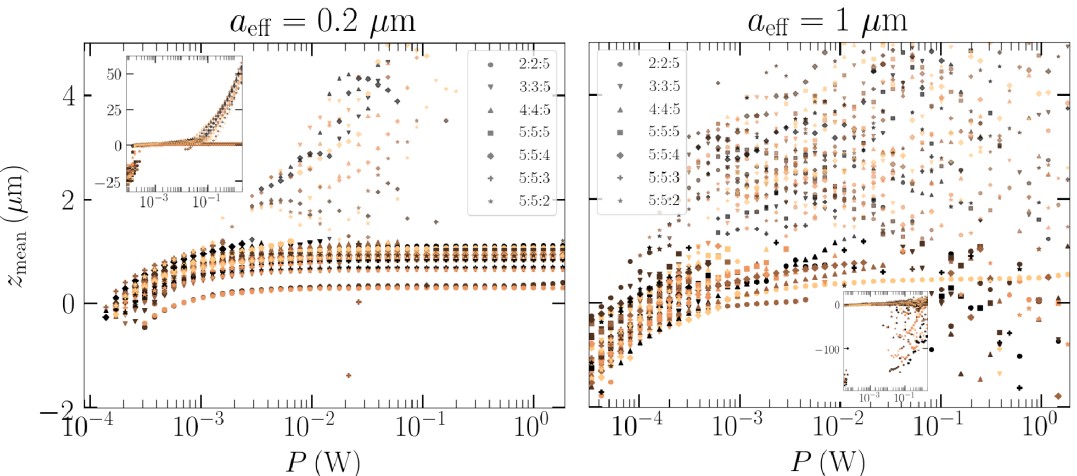

**Fig 10.** Left: The mean $z$-position for the Gaussian spheroid sample in air as a function of beam power for the 0.2 $\mu$m sample. The small Fig is a zoom-out of the same image, where particles in a beam too weak are falling due to gravity and particles ejected by the too strong beam are flying in the positive z-direction. Right: The same as left, but for the 1 $\mu$m sample.

$\mu$m. Each of the 35 particles has its response integrated using 50 separate beam powers ranging from 0.1 mW to 1 W.

First, in the left panel Fig 10, a general trapping is observed between 0.1 and 2 mW beam powers, where every particle regardless of shape can be trapped. The minimum beam power was determined for a random single particle from the sample, which results that the particles with smaller scattering cross section are not held in the trap. Moreover, the minimum power for ejection is much smaller for some particles than for the example Gaussian ellipsoid in Fig 8.

Second, in the right panel, the same test for the 1 $\mu$m sample is considered. There we observe a clearly more sensitive response to the beam power. Still, a considerable portion of the particles will hover stably just below the beam focus when the beam power is of the order 0.1 mW. The lower minimum trapping powers could easily be explained by the fact that the larger particles are wavelength-sized, and will intercept nearly all of the power available at the paraxial beam focus, at which point the width is, according to Fig 2, one wavelength.

Finally, a significant portion of the particles for which trapping is possible in the largest wavelength ranges are more elongated or sphere-like than flattened. This could be intuitively explained by a preferential rotational state, where the minor or major inertia axis is oriented parallel to the beam direction, and the angular momentum is also along the same direction. This would guarantee that the flattened shapes then have the maximum scattering cross-section, and will be ejected most efficiently.

The virtue of full dynamical integration allows us to test this directly. If a correlation of shape-dependent rotational state appears with the tendency to be trapped or ejected, the previous hypothesis would be upgraded to the primary explanation. We perform tests for the two particle samples in the case where trapping is successful, in beam power range 0.1 to 1 mW. In Fig 11, it is shown that regardless of shape, the angular velocity of the particle in a circularly polarized beam is parallel or antiparallel to the beam wave vector **k**. In such a well-behaving system, the so-called internal alignment, or the direction of angular momentum in the principal body coordinates is relatively simple to study. The azimuth and elevation of angular momentum in the body coordinates is nearly constant in all cases, and can be plotted directly. This reveals that a significant fraction of particles is not in the area corresponding to the

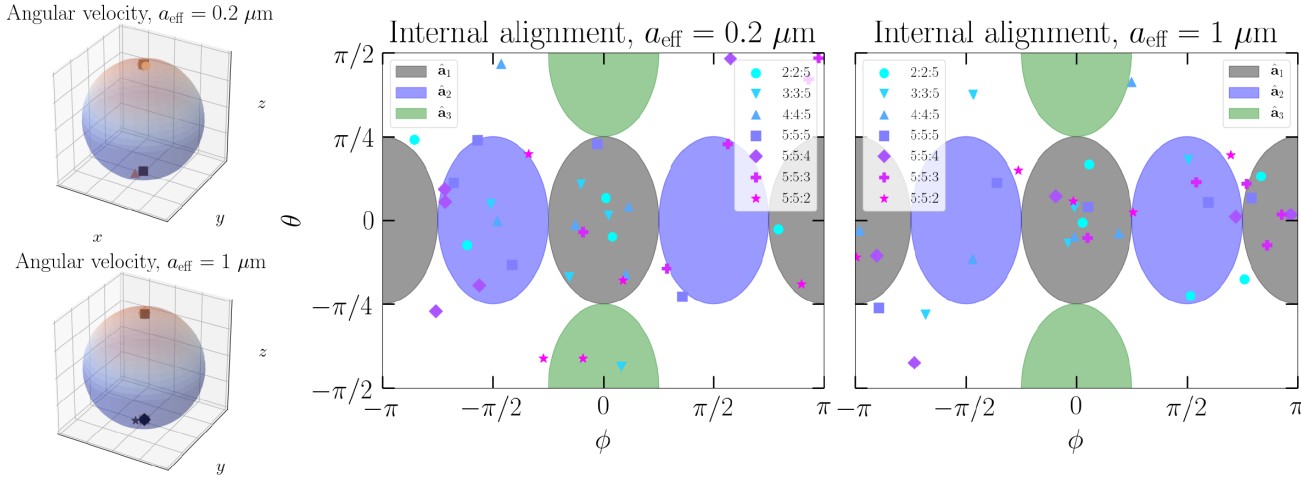

**Fig 11. The directions of angular velocity for the 0.2 and 1 $\mu$m samples in laboratory coordinates, where k ∥ z for the beam, and the corresponding internal alignment of angular momenta.** In body coordinates, the origin corresponds to the direction of principal axis of smallest moment of inertia (minor axis $\hat{\mathbf{a}}_1$, grey), and $(\phi, \theta) = (0, \pi/2)$ the direction of the major principal axis ($\hat{\mathbf{a}}_3$, green).

internal alignment with respect to the minor axis. Instead, many particles are found to be aligned with the intermediate axis or not strongly at all.

The alignment thus reveals that for these deformed spheroids, the naive hypothesis is a potential explanation for the shape-dependent behavior. For elongated particles, the intermediate axis is close to the minor axis in magnitude, and thus mean that the elongated particles are spinning 'standing upright', or along the geometric long axis. For flattened particles, such a state of rotation means that the cross section is nearly maximized, as they are spinning closely about their geometric short axis, or the major principal axis. Trapping using a single nonadaptive beam in vacuo would be much more difficult to achieve, as the drag of the medium provides some for the particles to reach a stable rotational state. In other types of beam polarizations, the helicity of deformed shapes can also affect the rotational response in a more complicated manner, making trapping more difficult.

### 3.3 Brownian motion

After testing how the particles tend to behave under drag, randomizing effects are important to consider. For example, in a dual-beam optical trap, Brownian motion in air has been observed for 3-micron silica beads [30]. In this section, we analyse the effect of Brownian motion on sub-micron silicate particles in the simulated single-beam LG00 trap.

In this section, we focus on air as the medium. This implies that the friction coefficient is two orders of magnitude smaller than in water, and the Brownian force subsequently is about an order of magnitude smaller than in water. However, drag effects are expected to become much more prevalent and resulting in more efficient trapping than in air. Due to this, and for the high relative computational cost, the following analysis is done for particles in air.

An initial demonstration on the implications of adding Brownian motion is presented in Fig 12, where a 0.2-$\mu$m particle is subjected to a 83-mW beam and Brownian motion is turned both off and on. The average behavior is initially the same, the particle stays at the z-position of 1 $\mu$m and near the x, y-origin. Eventually, Brownian motion forces the particle to escape the trap. As the beam power is not critically high in this case, the random motion determines whether or not the particle can return to the trap. In this particular case, the particle eventually

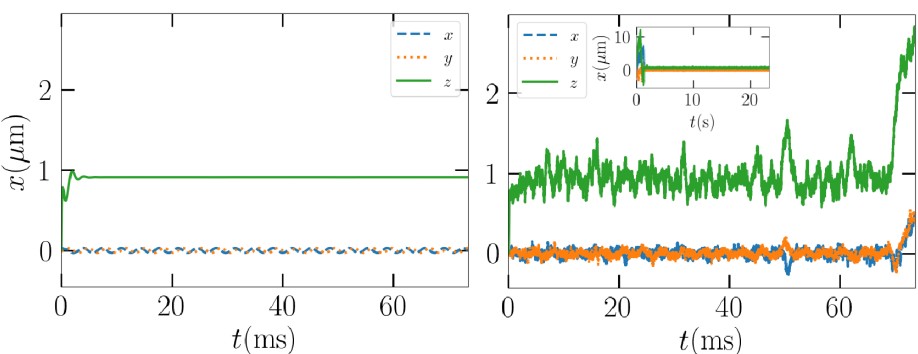

**Fig 12.** Left: The path of the 0.2 $\mu$m Gaussian ellipsoid example particle without Brownian motion. Right: The same with Brownian motion enabled with a zoom-out to of the whole simulation.

drifts back to the trap in a seemingly more stable rotational state, as it stays in the trap over 20 seconds or until the simulation was terminated.

While the particle stays on average at the same position as the unperturbed counterpart, variance in the $z$ direction is much greater than in the transversal $x$, $y$ directions. This can be explained by the general property of an upward beam, where scattering lowers the $z$-directed trap efficiency, which is also apparent in the left panel of Fig 3 for the cube.

Motivated by this, we probe the variance of the transversal position, or the $x$, $y$ distance from the origin, as a function of beam power, which affects the slope of the trap efficiency around the focus, also known as the spring constant $k$ of the trap, or trap stiffness. Pedantically, in this work, the trap efficiency is unitless and trap stiffness is often expressed in SI units, but they are interchangeable when the beam power is known.

In Fig 13, we see that as the beam power rises, the transversal variance decrease can be fitted by $1/k$. The trap stiffness range corresponds to a beam power range from 5 mW to 0.36 W for both the cube and the example Gaussian ellipsoid, for both of which the transversal trap efficiency slopes are very close to 12,500 1/m. In both cases, 1,000 separate beam powers were considered. For specific values of $k$, the particle transversal position is plotted over longer

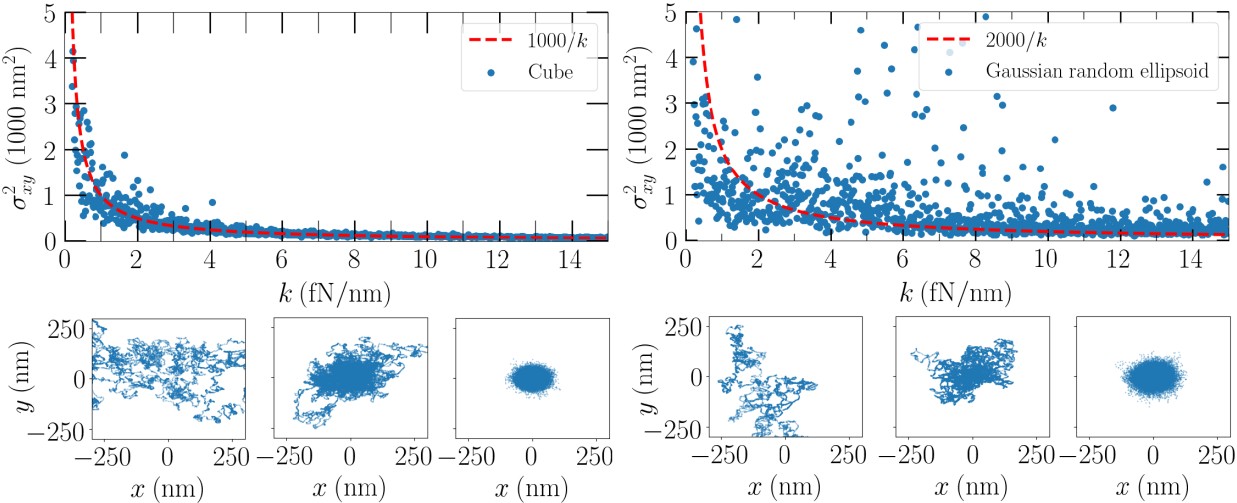

**Fig 13.** Top panels: Transversal $x$, $y$ distances from the origin for the cube and the example Gaussian ellipsoid, respectively. Bottom panels: Paths traced for the corresponding particles when $k$ = 0.3, 1.1, and 3.1 fN/nm.

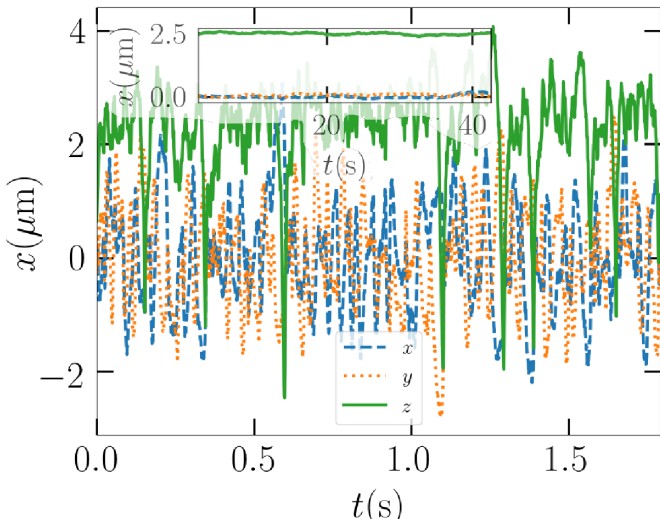

**Fig 14. The path of the example Gaussian ellipsoid of size 1 μm when Brownian forces are present.** The inset figure shows the rolling time average of the position over 40 seconds. Note the slightly transparent inset axis labels.

timespans. Namely, between values $k$ = 0.3, 1.1, and 5.1 fN/nm, a shift from visibly more or less free random walk to a trapped particle is observed for both the cube and the example Gaussian ellipsoid.

According to these results with respect to the results presented in Fig 10, not all particles are possible or simple to trap when Brownian forces are present, but those with high critical beam power before ejection should in general be relatively easy to trap. Larger particles should also be trapped even when Brownian forces are present, similarly as in an example case provided in Fig 14. In water, where drag forces grow linearly with viscosity, the Brownian forces are expected to disturb the situation less critically.

## 3.4 Different compositions

Until now, we have considered only quartz-like silicate particle compositions. For these particles, the total magnitude of refractive index is lower than of those considered in this section: diamond, a generic opaque material, and a metallic-like material with a significantly high absorptivity. However, in the scope of this article, it is not feasible to redo similar analysis for these three materials. Instead, we focus on analysing the trap efficiencies.

By first comparing the trap efficiencies for a single Gaussian ellipsoid, in Fig 15, we see that silicate has the optimal properties for trapping. The transversal trap efficiencies of 0.2 μm are comparable to other compositions than the metal, but only the trap efficiency of silicate in the $z$ direction is low enough in the vicinity of the beam focus to expect efficient trapping. In the 1-μm case, the distinction between materials is even greater, with the opaque and metallic particles being actually repelled from the trap focus in the transverse directions, as the optical spring constant is negative in those cases.

These results would imply that a significant portion of differently shaped specimen would tend to drift higher in the trap, until the transverse trap efficiency would not be able to keep the particle above the focus, without even considering the case of negative spring constant. Systems, where trapping is inefficient or nonexistent in even a single direction, are rarely built, and often multiple beams and different beam directions are used. Recently, results on

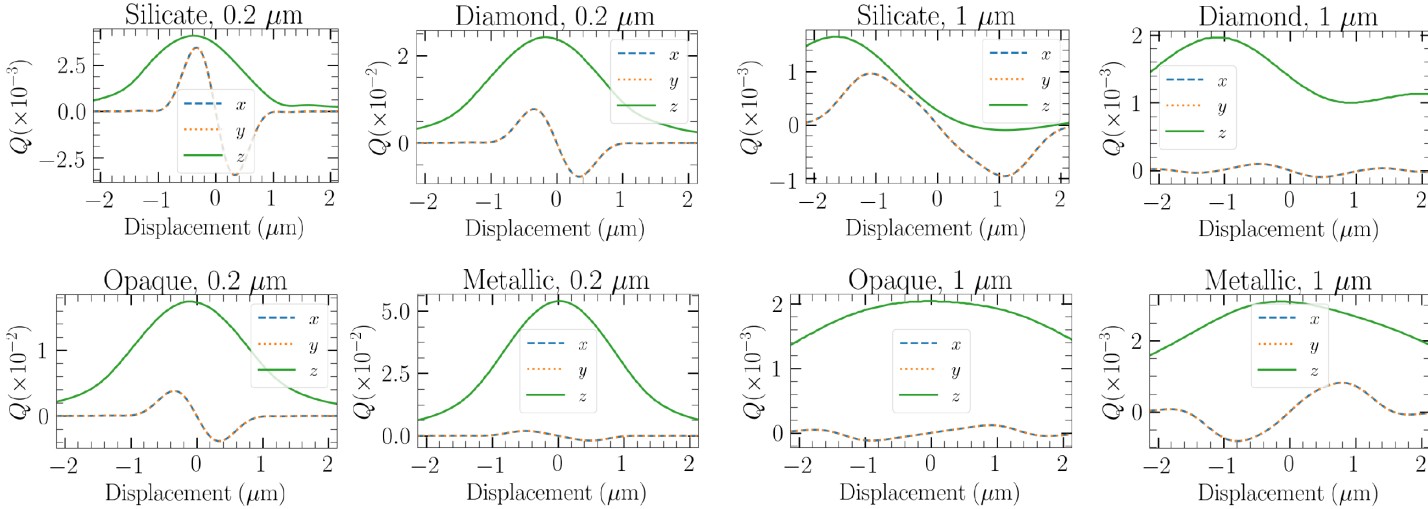

**Fig 15. Trap efficiency comparison for the example Gaussian ellipsoid particle of different sizes and compositions.**

manipulating wavelength-scale metallic particles with optical setups utilizing this have been published [31].

### 3.5 Different beams

As the last aspect of our study, we graze upon the central element of optical tweezers, the different beam types. A multitude of different beam types, from the $LG_{pl}$ modes ($p, l \neq 0$) reviewed in Section 2.2 via their Cartesian equivalent, the Hermite-Gaussian modes, to Bessel beams have been ignored thus far in this paper. In this section, using a single different beam type, the $LG_{03}$ mode with a hoop-like cross section, we demonstrate that the methods presented in this article can be applied to any given VSWF beam expansion, provided that the expansion is valid. This follows from the general property of the $T$-matrix formulation: the $T$-matrix does not depend on the electromagnetic fields.

In a circularly polarized $LG_{00}$ beam the particles will travel along the hoop-like beam intensity maximum. In Fig 16, we show the three differently sized Gaussian ellipsoids in the $LG_{03}$

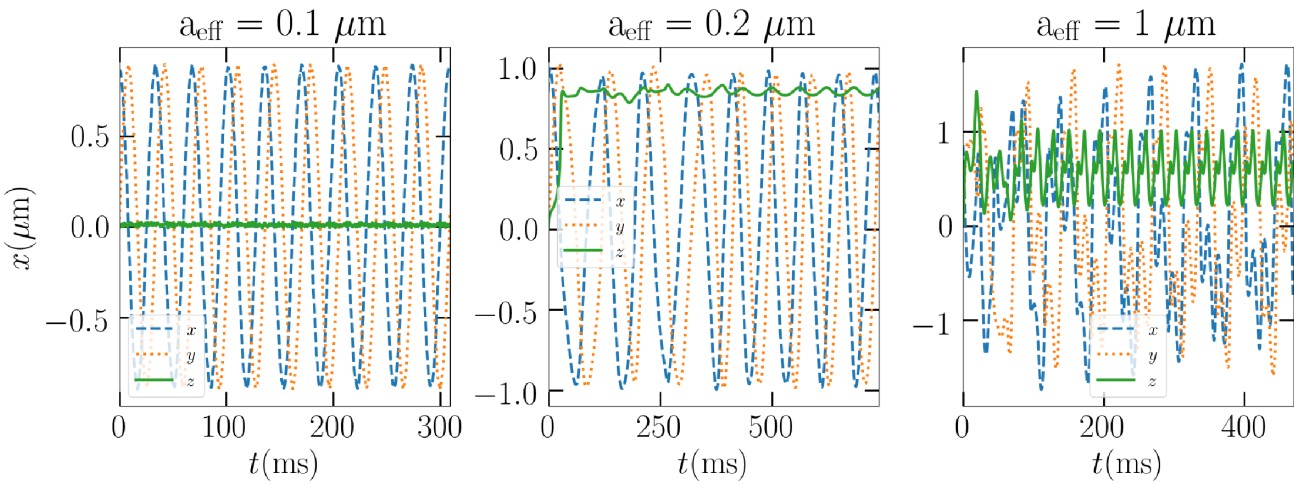

**Fig 16. The paths of the 0.1, 0.2, and 1 $\mu$m Gaussian ellipsoid particles in Laguerre-Gaussian ($LG_{03}$ mode) beam, respectively.**

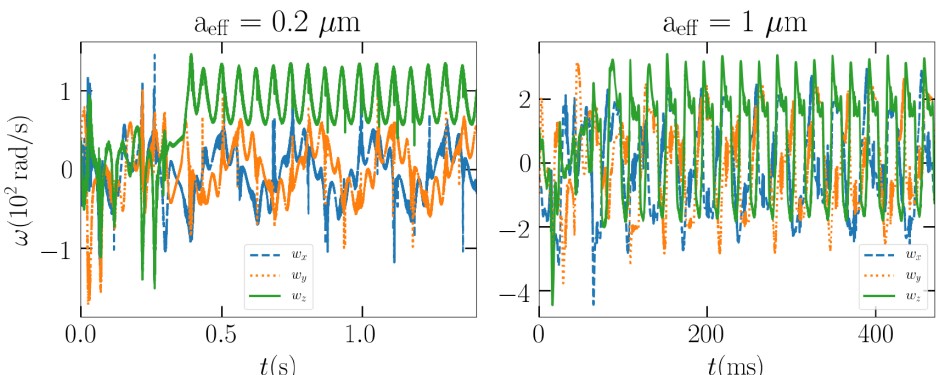

**Fig 17. The angular velocities in the laboratory frame of the 0.2 and 1-$\mu$m particles in the Laguerre-Gaussian (LG$_{03}$ mode) beam, respectively.**

optical tweezers, suspended in air. In all cases the beam power is 0.33 W. This power is chosen to be just enough for the beam intensity maximum to be able to counteract gravity (an electric field maximum of 8 MV/m). The trap efficiencies from earlier, and the simulations in the LG$_{00}$ beam, indicate that the smallest particle will stay trapped in the vacuum optical tweezers for the longest, with a possibility that the larger particles will fall.

We see that all the particles follow a circular path, even though in the 1-$\mu$m case it might not be apparent. The sporadic nature of the path is due to the particle rotating, thus its center of mass, which is graphed in the image, will move with a more complex period. We can be quite certain that the 0.2 and 1-$\mu$m particles truly move in a stable fashion by studying their angular momentum in the laboratory frame. In Fig 17, we see that the angular velocities quickly become periodic as well. In the case of the 0.1-$\mu$m particle, the angular velocity changes too sporadically for any stability arguments.

## 3.6 Summary

We have studied the simulation of optical tweezers by focusing on the aspect of possible media, shapes, compositions, and beam shapes. A special focus was given to the study of irregular shapes in optical tweezers, as the numerical method provides means efficient enough to solve scattering by arbitrarily shaped particles repeatedly. Combining the solution in a $T$-matrix form allows explicit integration of translational and rotational motion in electromagnetic fields, which coincidentally also is the very definition of an optical tweezers simulation problem.

The current study of single continuous beam tweezers can be summarized as such:

1. The choice of medium alters the particle response only little, if the change in trap efficiency is as moderate as in the test cases of this study. Zeros of the trap efficiency, and change therein, was seen to correlate with the final particle position. Otherwise, the motion and whether trapping is successful when the trapping is relatively inefficient is expected to be affected more by drag. In the case of no drag, trapping with a single-beam setup is much more difficult.

2. The particle shape has a complex relationship with the possibility of trapping. Still, the general shape of the particle can give some indication on how likely the particle is trapped, when the beam is known to force a certain angular velocity on the particles.

3. Interplay of Brownian forces and maximum allowed beam power before ejection complicates trapping significantly. However, a proper choice of medium can lead to more tolerance against Brownian effects due to the increase of drag and thus the maximum allowed beam power.

4. Trapping is highly dependent on the composition of the particles. Particularly, highly absorptive particles can be repelled from the focus, requiring much more complex beam setups to achieve trapping.

5. The advantage of a precalculated $T$-matrix is the ability to change the simulated beam at will. Not only are different beam VSWF expansions usable, also their superpositions can be used to simulate more complex beam setups.

## 4 Links to experiments

The tweezers beam can itself, especially when the beam shape is relatively simple, e.g. Gaussian, be used to measure the particle location and angular frequency [32]. Eq (1) provides the intensity profile of a Gaussian beam. A measurement of position and frequency as an inverse problem is illustrated in Fig 18. As the forward scattering direction is insensitive to particle shape, the relative intensity normalized to the scattered intensity of the particle at a random orientation at the origin can be used to estimate the distance to the initial position. Some periodicity in the inferred distance from the origin still appears, even though the real position vector is of constant length with some precession. This can likely be attributed to the irregularity of the shape, as the inferred distance has almost the same periodicity as the precession.

Similarly, scattered intensity in a different direction can reveal the rotational frequency, however it may not be immediately clear which frequency corresponds to a full revolution and which is caused by scattering fluctuations either due to the rotating irregular features of the particle or the precession of the particle. It should also be noted that such measurements are most useful when the particle stays near a fixed position and the incident beam is highly focused in the axis of propagation.

Measurements using the scattered beam are possible in principle, and can reveal details of both particle position and spin. However, the inclusion of Brownian effects would certainly make even the simulated measurement inversion much more difficult, if not impossible.

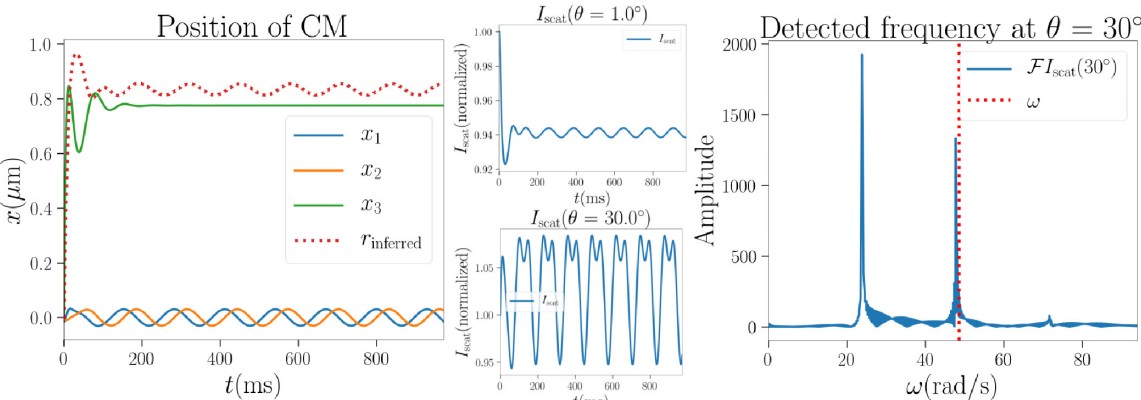

**Fig 18.** Left: Simulated position of the particle and simulated measurement of the distance from origin (dotted line). Middle: Simulated measurements of scattered intensity. Right: Simulated measurement of the angular frequency and true simulated final value (dashed line), where scattering fluctuations due to precession are the cause of the leftmost peak with lower frequency.

Actual measurements such as these would likely be feasible only if Brownian effects are small or when trapping is very efficient and counteracts Brownian motion.

## 5 Conclusions

We have presented a full solution to the dynamical problem of non-spherical particles illuminated by a continuous-wave Laguerre-Gaussian beam in a numerically exact manner. Both explicit integration of the equations motion or analysis of relevant dynamical measures, such as the trap efficiencies, can be used to obtain a picture of the mechanical effects of optical tweezers.

The focus of this study was to understand the response of differently shaped and sized particles in optical traps and tweezers. It was found that the general shape affects the optimal trapping beam power, but because the rotational state can be forced using a circularly polarized beam, general particle-shape-dependent trapping properties can be taken into account.

Specifically, a ground rule that circularly polarized beams tend to spin particles up in the intermediate principal axis direction, which nearly minimizes the scattering cross section of elongated particles and maximizes the scattering cross section of flattened particles, affecting the optimal beam power range of trapping. Explicitly, the range is larger for elongated particles than for flattened ones. Also, it was found that because of irregularities, an upper limit for the beam power, where particle is still able to find a stable rotational state before being out of the trapping zone, in contrast with highly symmetric shapes, was observed. Based on these observations, it may be beneficial to apply the numerical methods presented here to predict the usability of specific beam setups for trapping specific types of particles.

The framework presented in this work provides also the complete means of modeling arbitrary particles in optical tweezers with six degrees of freedom. Future improvements should consider e.g. the effect of particle deformations, particularly those of liquids: oscillating deformations or evaporation, the inclusion of geometric optics for larger particles [4], more detailed hydrodynamical simulations for particles in flows, or with other beam types, such as multiple-beam tweezers, different beam shapes, or pulsed lasers [33].

## Supporting information

**S1 Dataset. This dataset supplements this publication and contains inputs and processing scripts for usage of `scadyn`.** Also is contained the minimal dataset to reproduce the results presented in this article. Available at https://doi.org/10.5281/zenodo.3522835.
(ZIP)

**S1 Software. `scadyn`. A code for scattering dynamics calculations, which utilizes a volume integral equation solution for the $T$-matrices of non-spherical scatterers.** The main motivations for the development of this code are the study of grain alignment dynamics in interstellar environments, the study of and optical tweezers and traps. Available at http://www.github.com/jherrane/scadyn.
(ZIP)

## Author Contributions

**Conceptualization:** Johannes Markkanen, Gorden Videen, Karri Muinonen.

**Formal analysis:** Joonas Herranen.

**Funding acquisition:** Karri Muinonen.

**Investigation:** Joonas Herranen.

**Methodology:** Joonas Herranen, Johannes Markkanen.

**Resources:** Gorden Videen.

**Software:** Joonas Herranen, Johannes Markkanen.

**Supervision:** Karri Muinonen.

**Visualization:** Joonas Herranen.

**Writing – original draft:** Joonas Herranen.

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
