## [Decision Letter · Decision Letter 0]

21 Oct 2019

PONE-D-19-26831

Non-spherical particles in optical tweezers: a numerical solution

PLOS ONE

Dear Mr. Herranen,

Thank you for submitting your manuscript to PLOS ONE. After careful consideration, we feel that it has merit but does not fully meet PLOS ONE’s publication criteria as it currently stands. Therefore, we invite you to submit a revised version of the manuscript that addresses the points raised during the review process.

It is important that you address all the concerns raised by both the reviewers as I also believe it will certainly raise the value of the article.  Some references to the more recent work is required as suggested.

We would appreciate receiving your revised manuscript by Dec 05 2019 11:59PM. To enhance the reproducibility of your results, we recommend that if applicable you deposit your laboratory protocols in protocols.io, where a protocol can be assigned its own identifier (DOI) such that it can be cited independently in the future. For instructions see: http://journals.plos.org/plosone/s/submission-guidelines#loc-laboratory-protocols

We look forward to receiving your revised manuscript.

Kind regards,

Debabrata Goswami

Academic Editor

PLOS ONE

Additional Editor Comments (if provided):

I agree with the comments provided by both the reviewers and recommend that the authors appropriately address those and submit a revised manuscript for further consideration.

Journal Requirements:

Reviewers' comments:

Reviewer's Responses to Questions

**Comments to the Author**

1. Is the manuscript technically sound, and do the data support the conclusions?

Reviewer #1: Yes

Reviewer #2: Partly

2. Has the statistical analysis been performed appropriately and rigorously? 

Reviewer #1: Yes

Reviewer #2: N/A

3. Have the authors made all data underlying the findings in their manuscript fully available?

Reviewer #1: Yes

Reviewer #2: No

4. Is the manuscript presented in an intelligible fashion and written in standard English?

Reviewer #1: Yes

Reviewer #2: No

5. Review Comments to the Author

Reviewer #1: Most parts are well-written. I particularly enjoyed the concise way that many phenomenon were explained (e.g. Lines 166-170).

Major point:

1. All of the examples are in the size parameter range of about ka=1 (and up to ka=5.9 later). Are there numerical problems that don't allow larger size parameters? For instance ka=40 to 100 would be relevant to many experiments. It is hard to tell how these results scale.

2. Line 84: "In thought experiments, two types of forces, scattering and gradient force, are usually considered separately." This sentence comes across as very dismissive to decades of work on modelling optical trapping. Please review that work properly and discuss the motivation behind separating the forces if you are going to mention it.

3. When moving from air to water the effect of the medium on the relative refractive index and, subsequently, trapping efficiency is usually discussed. Unless I missed it, such a discussion is absent here.

Minor points:

1. Fig. 3, left-panel: Are these the components of Q_F or Q_N?

2. Line 299 and Fig. 10 caption: Why does k have a hat on it? Is it a unit vector?

3. Fig. 12 caption: No units for k?

4. Fig. 13: What is going on with that inset? Very odd figure construction.

Reviewer #2: Please check the attached review. There are copyediting and clarifications which will be needed. The data-collection policy must be clarified further. The work itself addresses an interesting problem in a coherent way.

6. PLOS authors have the option to publish the peer review history of their article (what does this mean?). If published, this will include your full peer review and any attached files.

Reviewer #1: No

Reviewer #2: No

---

## [Author Response · Author response to Decision Letter 0]

1 Nov 2019

Response to Reviewers

The authors thank the anonymous reviewers for their comments. The comments were fruitful, and resulted in the addition of over 2 pages to the manuscript. Many of the comments also provide relevant ideas on possible applications of the methods to currently undergoing experimental studies. The discussion on the drag terms is reworked independently for consistency and to completely describe what is needed to be implemented numerically. The response to each point raised is in the following.

Review #1

Major point:

1. All of the examples are in the size parameter range of about ka=1 (and up to ka=5.9 later). Are there numerical problems that don't allow larger size parameters? For instance ka=40 to 100 would be relevant to many experiments. It is hard to tell how these results scale.

The answer in many ways is implicit to the T-matrix method itself, which is rarely re-reported in

numerical application studies. Short answer is that the results do not scale in an obvious manner.

The size regime is in the highly non-trivial transition between Rayleigh scattering and geometric optics. The convergence to geometric optics may be possible to probe using the T-matrix method, but with considerable computer resources and far less different or far more constrained particle shapes (utilizing different methods than in this work).

The T-matrix method gives a numerically exact solution in the Mie regime of scattering, where both Rayleigh scattering and geometric optics are expected (and demonstrated) to fail. The T-matrix is a full square matrix of size O(ka^2 * ka^2), making even the fastest current numerical methods vastly outperformed by the Mie solution (for spheres, easily reaching size parameters up to 100 and higher). This practically restricts the upper limit of ka to less than 20, and for decent performance on non-cluster environment, such as in this work, the upper limit is absolutely advised to be under 10.

2. Line 84: "In thought experiments, two types of forces, scattering and gradient force, are usually considered separately." This sentence comes across as very dismissive to decades of work on modelling optical trapping. Please review that work properly and discuss the motivation behind separating the forces if you are going to mention it.

Rewording has been done. The original message of this part was not to be dismissive to any previous work, but to reason the omission of the canonical approach to describe optical tweezers. By explicitly mentioning that analytic decomposition of the forces can only be done for Mie scattering in the Mie size regime, this message should be more difficult to read as dismissive. For cases such as in this work, an analytic decomposition does not exist (dx.doi.org/10.1103/PhysRevA.100.033821 for the most recent work on this question found at this time, again done for Mie spheres, not non-spherical scatterers), thus the total optical force is only considered.

3. When moving from air to water the effect of the medium on the relative refractive index and, subsequently, trapping efficiency is usually discussed. Unless I missed it, such a discussion is absent here.

Trapping efficiency discussion is added to section 3.1 along with a new Fig 5.

Minor points:

1. Fig. 3, left-panel: Are these the components of Q_F or Q_N?

Forgotten Q_F added to the figure.

2. Line 299 and Fig. 10 caption: Why does k have a hat on it? Is it a unit vector?

Answer to last question: yes. However, as there is no particular need to choose one over the other for conveying the physical point, so changes to both were done to simplify notation. (Now Fig 11)

3. Fig. 12 caption: No units for k?

Units added, also in-text. (Now Fig 13)

4. Fig. 13: What is going on with that inset? Very odd figure construction.

Similar approach is used in Fig 6., where it is crucial to see the effects in different time scales. In Fig 14 (previously Fig 13), it is explicitly stated that a rolling average of particle position is used, with the implicit reasoning that it shows that the bouncing motion will continue for a very long time. If averaging was not done, the figure would be an incomprehensible mess of colours. The problem with the image is a rather subjective compromise to fit correct axis labels without covering much of the plots, and as such the inset labels are transparent.

Review #2

Grammar and Spelling

All points were reviewed and applied during the second proof-reading with the exception of points of lines 8, 22, and 53, where the correction suggestion was incorrect.

The points below are indicative and non-exhaustive.

7: "are considered, such as in [3]" -> "are considered [3]"

• Revised accordingly

7: "due to that dynamical modeling of the" -> "due to the fact that the

dynamical modeling of"

• Revised accordingly

8: "tweezers requires" -> "tweezers require"

• “Requires” refers to a singular form earlier in the sentence, no revision

10: "for considering" -> "to solve for"

• Revised accordingly

11: "Frameworks" -> "Existing frameworks deal with"

• Revised accordingly

15: "has fittingly" -> "has previously been"

• Removed fittingly

16: "about simulating" -> "related to the simulation of"

• Revised accordingly

20: "methodology" -> "methodology presented"

• Revised accordingly

21: "exact volume integral equation solution" -> "exact solution of the

volumen integral equation"

• Revised accordingly

22: "is indifferent" -> "is insensitive"

• Indifferent and insensitive are not interchangeable, so rewording was done to make the message unambiguous.

27: "on a category" -> "on the category"

• Revised accordingly

29-30: "we introduce a numerical.." -> "we introduce a model for numerical

studies of optical tweezers for arbitarily shaped rigid particles. These have

been shown experimentally"

• Revised

47: "as a rigid tetrahedral mesh" -> "as rigid tetrahedral meshes"

• Revised accordingly

51: "nor geometric" -> "nor the assumptions of geometric"

• Revised accordingly

53: "deformed ellipsoidal" -> "deformed ellipsoidal particles"

• Revised the whole sentence

58: "is in" -> "for"

• Revised accordingly

69: "indeed require careful adoption" -> "need to be adapted carefully"

• Revised accordingly

98: "In optical tweezers," -> "When considering classical optical tweezer

theory,"

• Revised accordingly

130: "forcing on" -> "forces acting on"

• Revised accordingly

130: "tweezers is" -> "tweezers are"

• Revised accordingly

137: "forcing" -> "forces"

• Revised accordingly

138: "the exact" -> "this exact"

• Revised accordingly

139: "the Brownian force is the" -> "the Brownian force is accounted for by

the"

• Revised accordingly

142: "force direction" -> "force vector"

• Revised accordingly

Clarifications

• 8: "repeated solution" is unclear. Do the authors mean a recursive solution is required?

◦ Paragraph is modified to describe the problem at hand more precisely.

• 91: Citation required

◦ Citations added

• 103: Citation required

◦ The paragraph is expanded considerably. As to the citation, it is unclear what to cite, as the phrase is seemingly as true as stating that electromagnetic scattering of arbitrary shapes is difficult or impossible to solve without numerical methods.

• 106: The mechanical interactions are unclear.

◦ Rewording to clarify that interactions are momentum-transferring.

• 103-104: The authors have not given enough background on the background of the analytic spherical particle results used, and the rational for accounting for the macroscopic flow properties in terms of their order-of-magnitude estimates should be described further.

◦ The drag terms on spherical particles are already described in later equations (3)-(6). The rationale for accounting these terms is added to the third paragraph of section 2.3.

• 126-131: Citations required

◦ Citation and self-reference added.

• 140-141: The reference, [22] mentions "Given that We will consider a silica microparticle in water with radius R = 1 μm, mass m = 11 pg, viscosity η=0.001 Ns/m2, γ=6πηR, temperature T = 300 K, and τ=0.6 μs. We remark that τ is orders of magnitude smaller than the time scales of typical experiments" The authors should comment on how the differences in the systems considered are accounted for.

◦ Some rewording. The approach is now more clearly described in a way that the relation to the cited paper is more clear (inertial ballistic assumptions hold elsewhere, as the inertial part is never disregarded in this work).

• 150-154: It is unclear how the Brownian forces are taken into account and the discussion should be left to section 3.3.

◦ A link to the later section is added. The distinction between theory and simulation parts of the article is enforced quite strictly elsewhere, but as a such short section, 2.3.1 benefits from the mention of the general approach of the paper, as it helps to justify the existence of this short section.

• The trapping detail does not cover if situations where multiple particles enter the trap, and the orientation and scattering should be tested against experimental studies [Mondal, Dipankar, Anushka Jha, Yogesh M. Joshi, and Debabrata Goswami. "Microrheology Study of Aqueous Suspensions of Laponite using Femtosecond Optical Tweezers." In Optical Trapping Applications, pp. OtW2E-1. Optical Society of America, 2017.]

◦ The reworked 3rd paragraph of section 2.3 provides some constraining arguments on this issue. This point is difficult to address further in-text, as it is by any means untouchable by the T-matrix method, if any electromagnetic interaction between the particles is assumed. Also, as the collisions between particles are disregarded, and flow dynamics is implicitly only flow-on-particle effects, not particle-on-flow, there would be no meaning in considering multiple particles in the scope of this work.

◦ The scattering solution is cited properly to articles by Markkanen&Yuffa, Markkanen et. al., and Herranen et. al., along with Farsund&Felderhof and Crichton&Marston for the analytic relation between a numerical scattering method and forces and torques. The methods are all tested against other numerical methods, that have had time for extensive verifications, both numerical and experimental.

• 318-331: Implementation details of the Brownian motion should be added, along with where it ties into the data made available. References are required to show significant Brownian motion in air, being as it is primarily a liquid-state effect.

◦ The Brownian motion is implemented as described in the manuscript, as an Langevinian addition to the total force used in the integration scheme, which is described in an earlier article and also available in the source code. Reference added.

• 381-387: Given that the VSWF expansion has not been shown mathematically for different beam types this section should be reworked to describe that the results are heuristically determined from the numerical simulations.

◦ References to the VSWF expansions used in this work (described by Nieminen et. at.) are given in section 2.2 (part of theory), and results provided in section 3.5 (part of simulations) are not heuristic. The conclusion that the methods can be used, if a VSWF expansion can be given, is reworked into the text, the shred any hints of heuristic claims.

• Section 4 needs references from the existing literature.

◦ Reference added to section 4, describing an experiment very similar to the simulated setup.

General comments:

1. The authors should mention that the optical tweezers considered are for continuous wave (CW) lasers and perhaps add to how they expect pulsed laser systems to behave. The work of [De, Arijit Kumar, Debjit Roy, Aveek Dutta, and Debabrata Goswami. "Stable optical trapping of latex nanoparticles with ultrashort pulsed illumination." Applied Optics 48, no. 31 (2009): G33-G37] and [Deng, Jian-liao, Qing Wei, Yu-zhu Wang, and Yong-qing Li. "Numerical modeling of optical levitation and trapping of the “stuck” particles with a pulsed optical tweezers." Optics express 13, no. 10 (2005): 3673-3680] should be mentioned in terms of pulsed lasers.

This was due to non-standard usage of terminology in this manuscript. Continuous wave lasers were referred to as static beams in the abstract. The term CW is now adopted in the abstract and conclusions section. The consideration of pulsed beams however is out of the scope of this article, as are many other types of tweezers. Pulsed beam system was added as an example of Conclusions section with relevant citation.

2. The data availability is slightly unclear. The software itself is located at the Github repository and with the appropriate license, however the data used for this study is not evidently a part of this manuscript or the supporting materials. Please clarify this.

The minimal dataset (PLOS definition) will be added to the Zenodo repository.

---

## [Decision Letter · Decision Letter 1]

13 Nov 2019

Non-spherical particles in optical tweezers: a numerical solution

PONE-D-19-26831R1

Dear Dr. Herranen,

We are pleased to inform you that your manuscript has been judged scientifically suitable for publication and will be formally accepted for publication once it complies with all outstanding technical requirements.

With kind regards,

Debabrata Goswami

Academic Editor

PLOS ONE

Additional Editor Comments (optional):

Reviewers' comments:

Reviewer's Responses to Questions

**Comments to the Author**

1. If the authors have adequately addressed your comments raised in a previous round of review and you feel that this manuscript is now acceptable for publication, you may indicate that here to bypass the “Comments to the Author” section, enter your conflict of interest statement in the “Confidential to Editor” section, and submit your "Accept" recommendation.

Reviewer #1: All comments have been addressed

Reviewer #2: All comments have been addressed

2. Is the manuscript technically sound, and do the data support the conclusions?

Reviewer #1: Yes

Reviewer #2: Yes

3. Has the statistical analysis been performed appropriately and rigorously? 

Reviewer #1: N/A

Reviewer #2: Yes

4. Have the authors made all data underlying the findings in their manuscript fully available?

Reviewer #1: Yes

Reviewer #2: Yes

5. Is the manuscript presented in an intelligible fashion and written in standard English?

Reviewer #1: Yes

Reviewer #2: Yes

6. Review Comments to the Author

Reviewer #1: (No Response)

Reviewer #2: I would like to thank the authors for addressing all the comments satisfactorily, and I am very pleased to recommend the revised manuscript for acceptance in its present form.

7. PLOS authors have the option to publish the peer review history of their article (what does this mean?). If published, this will include your full peer review and any attached files.

Reviewer #1: No

Reviewer #2: No

---

## [Editor Report · Acceptance letter]

19 Nov 2019

PONE-D-19-26831R1

Non-spherical particles in optical tweezers: a numerical solution

Dear Dr. Herranen:

I am pleased to inform you that your manuscript has been deemed suitable for publication in PLOS ONE. Congratulations! Your manuscript is now with our production department.

With kind regards,

on behalf of

Dr. Debabrata Goswami

Academic Editor

PLOS ONE